# Research on the Choice Behavior of Taxis and Express Services Based on the SEM-Logit Model

**Yang Si *** **, Hongzhi Guan and Yuchao Cui**

College of Architecture and Civil Engineering, Beijing Key Laboratory of Transportation Engineering, Beijing University of Technology, Beijing 100124, China; hguan@bjut.edu.cn (H.G.); cuiyuchao@emails.bjut.edu.cn (Y.C.)
* Correspondence: siyang@emails.bjut.edu.cn

**Abstract:** With the development of Internet technology, online car-hailing is booming in China, which has profoundly affected people's travel structures. In order to seek the sustainable development of taxi and online car-hailing services from the perspective of passenger mode choice behavior, the mechanism of passengers' decision-making procedures and their travel mode choice behaviors were analyzed. To study the influence of latent variable factors on passenger choice behavior, this paper firstly designed a questionnaire, and a structural equation model (SEM) was established for the preliminary study of the relationship between the latent variables and the behavioral intentions using the online survey data. Then, the latent variables were introduced into the Logit model, setting up the SEM-Logit model to explore the mode choice patterns between taxis and online car services. The results showed that the SEM-Logit model with the latent variables is better than a general Logit model in terms of the model precision and hit ratio. Meanwhile, after introducing the latent variables, it was found that convenience, comfort, and economy factors have a significant influence on the model, and the explanatory power of the model increases accordingly.

**Keywords:** online car-hailing; taxi; trip mode choice behavior; structural equation model (SEM)

---

## 1. Introduction

The trip mode choice is important in traffic demand analysis. Most factors that affect the trip mode choice can be directly observed, such as required cost, time, and other economic indicators, and have been widely applied in various research papers. In addition, it is also believed that there are some potential or non-directly observable factors, such as attitudes and feelings.

In recent years, online car-hailing has become popular in China, and a number of ride-hailing platforms have formed as a consequence. Online car-hailing services realize the information required for matching of the driver and the passenger and reduce the proportion of customers searching. Meanwhile, it can replace the private car travel needs of some high time value groups. Hence, it is a sustainable travel mode. Unlike taxis, online car-hailing services can only carry passengers through network appointments and are not allowed to cruise (Service types of them are showed in Figure 1). Although both online car-hailing services and taxis can provide personalized and door-to-door services (both of them are on-demand mobility services, and taxis additionally can be used for street hailing [1]), some researchers have shown that there are certain differences in the travel characteristics between them. Cui et al. [2] analyzed the online order data of taxis and express services (one type of online car-hailing service; the service and the price are roughly equal to a taxi) and found that, in terms of the travel time of taxi online orders, trips of less than 20 minutes account for 22.8% of all rides, trips of 20–30 minutes account for 24.5%, and trips of 30–40 minutes account for 21.2%. Moreover, the hot spots for taking on or off vary greatly. However, the online order data of express services are

concentrated on rides of 10–20 minutes, and 66.7% of them are within 20 minutes. Furthermore, the hotspot areas for taking on and off are basically consistent. From the service-indicator characteristics of online car-hailing services and taxis, the differences between them are not as obvious as between taxis and subways or taxis and buses. In these cases, passengers are able to distinguish them clearly, indicating that there are some personal choice preferences between the two modes that are difficult to describe with observable indicators. These factors affect the passengers' mode choice attitudes, which in turn influence the results. The purpose of this paper was to explore how potential factors (latent variables, such as the attitudes or the feelings of passengers) affect the choice behaviors for taxis and online car-hailing.

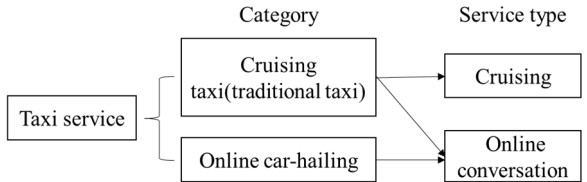

**Figure 1.** Taxi service classification.

The rapid development of online car-hailing has had a great impact on the taxi market, because it has realized information matching between passengers and drivers by using Internet technology. Isaa and Davis (2014) [3] insisted Uber severely disrupted the taxi service industry. Rayle et al. (2015) [4] investigated the usage of online car-hailing in San Francisco, USA, and explored the characteristics and the reasons of online car-hailing users as well as the influence of the online car-hailing market on the traditional taxi market, public transport, and private cars. The research suggested that online car-hailing can be a substitute for public transport and private cars to some extent. McKenzie and Baéz (2016) [5] differentiated Uber and taxi transportation through events attended by their passengers and explored event detection at a variety of spatial and temporal resolutions. Harding et al. (2016) [6] discussed taxi apps and their impact on taxi markets and suggested regulators should focus on reducing the likelihood of monopoly and collusion in a taxi market led by apps. Flores and Rayle (2017) [7] explained how and why Uber came to be accepted in San Francisco. Jiang et al. (2018) [8] comprehensively compared Uber, Lyft, and taxis with respect to key market features. Watanabe (2016) [9] considered that Uber used a disruptive business model driven by digital technology to trigger a ride-sharing revolution and fundamentally improved the efficiency of the load factor. Kim (2018) [10] concluded that Uber and taxis can create positive values through well-intentioned competition.

In terms of the travel characteristics analysis of online car-hailing, Dawes (2016) [11] found that there are relationships between the transportation characteristics of Uber or Lyft and user identification and attitude. Poulsen et al. (2016) [12] conducted a geo-spatial analysis of Green cab and Uber rides in the outer neighborhoods of New York City and found that demand for Green cabs is still growing, but that the number of Uber rides in the same area is growing more rapidly, while Green cabs are performing better than Uber in relatively poor neighborhoods. However, when looking at differences between weekdays and weekends, they found no differences between Green cabs and Uber. Abel and Kerry (2018) [13] showed that the number of Uber and Lyft rides is significantly correlated with whether it is raining by using all taxi, Lyft, and Uber rides in New York City. He and Shen (2015) [14] proposed a spatial equilibrium model that balanced the supply and the demand of taxi services in a regulated taxi market with the smartphone-based e-hailing application. Brodeur and Nield (2016) [15] found that the number of Uber rides per hour increases when it is raining, and surge pricing encourages an increase in supply. During the same time period, however, the number of taxi rides per hour decreased after Uber entered the New York market. Chen (2016) [16] studied driver work practices under the dynamic price markup of Uber and found that the driver's work adjustment is more flexible when the premium is high.

In the research on the choice behavior of online car-hailing, Peng et al. (2014) [17] adopted the planning behavior theory (TPB), the rational behavior theory (TRA), and the technology acceptance model (TAM), and analyzed the behavior intentions of passengers using car-hailing software. The results showed that the perceived usability, the perceived usefulness, and the compatibility have positive indirect effects on user attitude, consequently affecting the user intention; subjective norms have positive direct impacts on user behavior intention, whereas perceived risk has a negative direct impact on behavior intention. The perceived price level has an impact on behavior intention and user attitude. Zhang (2017) [18] used the Push-Pull-Mooring (PPM) model to study the transfer intention of online car-hailing users. Zhu (2017) [19] constructed a passenger satisfaction model based on the structural equation model. Zhang et al. (2016) [20] set up a binomial logit model to analyze the urban residents' choice behaviors for taxi and online car-hailing. It is observed that scholars often consider the subjective attitudes of users in the study of behavior intention. In the 1970s, Spear (1976) [21] discussed how abstract transportation system characteristics can be quantified and included as explanatory variables in models of travel demand behavior. Since then, perceptions, feelings, preferences, and other variables have been taken into account in choice models, and it has been proven that these potential variables have a significant impact on the results of choice [22–24].

To sum up, it is not only the direct observation variables—such as passengers' personal characteristics and trip characteristics—that affect their choice behavior, but also many psychological factors (for example, personal feelings, attitudes, etc.) that cannot be directly observed. This paper established a choice model considering the latent variables. Firstly, the questionnaire data was sorted, the decision-making process of the trip mode choice behaviors of passengers was analyzed, and the appropriate variables were selected. The structural equation model (SEM) was adopted to discuss the influence relationships among the variables. Subsequently, the analyzed latent variables were substituted in the choice model, and the choice model with the latent variables included was constructed to explain and predict the choice behaviors of passengers. For the theoretical aspect, this paper conducted a model analysis of the selection behavior of taxi drivers and passengers at the psychological level, and a more suitable mechanism for the analysis of the behavior selection model was put forward. In practice, survey data were used for calibrating the model, which could provide suggestions for relevant departments to develop sustainable development policies for taxis and on-line car hailing, optimize scheduling distribution, and improve passenger satisfaction.

The structure of the paper is as follows. Section 2 outlines the model and the selected variables. Section 3 describes the data collection and the data statistics. Section 4 analyzes the results. Section 5 gives the conclusion.

## 2. Methodology

### 2.1. SEM-Logit Model

In order to describe the influence of subjective factors on the passengers' decision-making processes, a SEM-Logit model considering the latent variables was constructed. The model consisted of two parts—the first part was the SEM model, which was mainly used to describe the causal relationship between the latent variables of travel mode selection and the corresponding observation variables. The second part was the Logit model, which was used to express the nonlinear function relationship between the probability of choosing a certain travel plan and the potential variables influencing the decision. Research on the application of the SEM-Logit model includes Yáñez et al. (2010) [25], who used hybrid discrete choice models for an urban multimode choice case incorporating latent variables. Chen and Li (2017) [26] presented a mode choice model for public transport, which integrated the structural equation model and the discrete choice model with categorized latent variables. Ding et al. (2018) [27] investigated the influences of the built environment on car ownership and travel mode choice simultaneously by making use of a multilevel integrated MNL (multinomial logit model) and SEM. Han et al. (2018) [28] used SEM-NL methodology to explore the possible causal relationships

among personal waiting behavior, attitudes to bus service satisfaction, and travel mode choices of passengers waiting at a bus station. The specific description of the model is as follows:

1.  Improvement of the utility function

The latent variable is added to the fixed utility term so that the utility function includes not only explicit variables such as trip characteristics and personal socio-economic characteristics of passengers but also latent variables such as perceptions, attitudes, and so forth. The improved utility function can be expressed as [25]:

$$V_{in} = \sum_l a'_{il} s_{iln} + \sum_q b'_{iq} z_{iqn} + \sum_k c'_{ik} \eta_{ikn} \tag{1}$$

where $i$ refers to an alternative, $n$ is the number of passengers, $l$ is the number of directly observable characteristics for the passengers, $q$ is the number of directly observable characteristics for the trip mode, $k$ is the number of latent variables, $s_{iln}$ is manifest variables of personal characteristics, $z_{iqn}$ is manifest variables of the trip mode, $\eta_{ikn}$ is latent variables, and $a'_{il}$, $b'_{iq}$ and $c'_{ik}$ are parameters to be estimated.

2.  Adaptation coefficient calculation for latent variables $\eta_{ikn}$

In order to determine the adaptation coefficient of the latent variables, SEM is needed to describe the relationship between the latent variable and its measurement variable. The latent variables $\eta_{ikn}$ can be described by a series of corresponding measurement variables $x_{irn}$. Taking an external latent variable in the model as an example, for the observation model of the structural equation model of the attitude perception attribute in the trip mode selection of passengers, the vector form is expressed as:

$$\begin{pmatrix} x_{11} \\ x_{12} \\ \dots \\ x_{1n} \end{pmatrix} = \begin{pmatrix} \Lambda_{x1} \\ \Lambda_{x2} \\ \dots \\ \Lambda_{xn} \end{pmatrix} \eta_1 \tag{2}$$

The load factors (path coefficients) $\Lambda_{x1}, \Lambda_{x2}, \dots, \Lambda_{xn}$ explained by exogenous potential variable $\eta_1$ are regarded as the weight of each observation variable (index variable), and then the load factors are standardized; $\alpha_{x1}, \alpha_{x2}, \dots, \alpha_{xn}$ are used to represent the assigned weights.

$$\begin{cases} \alpha_{x1} = \frac{\Lambda_{x1}}{\Lambda_{x1} + \Lambda_{x2} + \dots + \Lambda_{xn}} \\ \alpha_{x2} = \frac{\Lambda_{x2}}{\Lambda_{x1} + \Lambda_{x2} + \dots + \Lambda_{xn}} \\ \alpha_{xn} = \frac{\Lambda_{xn}}{\Lambda_{x1} + \Lambda_{x2} + \dots + \Lambda_{xn}} \end{cases} \tag{3}$$

Finally, the survey values of the observed variables are substituted into the formula, and then the adaptation values of the potential characteristic variables in the attitude perception attribute for the passengers can be obtained.

$$\eta_1 = \alpha_{x1} x_{11} + \alpha_{x2} x_{12} + \dots + \alpha_{xn} x_{1n} \tag{4}$$

3.  Discrete choice model

To describe the decision-making behaviors of the passengers, a binomial variable $d_{in}$ (the value can only be 0 or 1 when $d_{in} = 0$, indicating that option $i$ is not selected; when $d_{in} = 1$, it indicates that option $i$ is selected) is introduced. The formula is shown as follows:

$$d_{in} = \begin{cases} 1 \ if \ U_{in} \geq U_{jn} \\ 0 \ otherwise \end{cases} \tag{5}$$

## 2.2. Model Specification and Hypothesis

To study the passengers' choice behaviors for taxis and online car-hailing, an online survey using a questionnaire was conducted to obtain research data. The SEM-Logit model was established by using the SEM combined with the Logit model. The model consisted of two parts—the first part was the SEM model, which was mainly used to describe the causal relationship between the latent variables of the trip mode choice and their corresponding observation variables, and between the latent variables and explicit variables. According to the literature [26,29–32], five types of exogenous latent variables—convenience, reliability, comfort, safety, and economy—were selected. The perceptual value and the behavioral intention were endogenous latent variables, and the specific explanation of the selected variables is shown in Table 1. The second part was the Logit model, which was used to describe the functional relationship between the probability of choosing a trip mode and the latent variables and explicit variables that affect the decision-making. It is worth noting that the observed variables had no influence on the individual choice behavior, and could only be used to measure the latent variables.

**Table 1.** Item design of the study variables.

| Number | Latent Variables | Measured Variables | |
|---|---|---|---|
| 1 | Convenience (TC) | TC1 | Flexible route |
| | | TC2 | Easy to hail car |
| | | TC3 | Convenient to transfer with other transportation |
| | | TC4 | Convenient to make payment |
| 2 | Safety (TS) | TS1 | Personal life is safe when choosing this mode |
| | | TS2 | Personal property is safe when choosing this mode |
| | | TS3 | The report of safety accidents in this mode has little influence on the choice |
| | | TS4 | Trusted security management of this mode |
| | | TS5 | Not worried about leakage of personal information |
| 3 | Reliability (TR) | TR1 | Waiting time is close to the expected |
| | | TR2 | Arrive on time |
| 4 | Comfort (CC) | CC1 | Stable running status of vehicle |
| | | CC2 | Good vehicle condition |
| | | CC3 | Comfortable interior environment of vehicle |
| | | CC4 | Friendly driving attitude of driver |
| 5 | Economy (TE) | TE1 | Reasonable price |
| | | TE2 | Normative ticket system |
| 6 | Perceived Value (PV) | PV1 | High cost performance |
| | | PV2 | Satisfying service level |
| 7 | Behavioral Intention (BI) | BI1 | Be willing to choose this mode in the next year |
| | | BI2 | Happy to recommend this mode to others |

Before the establishment of the model, the following assumptions needed to be made regarding the model: (1) travelers were rational in making mode choices, as they would choose the travel scheme with the highest utility value; (2) options of mode choice were categorized into two types, taxis and express services; (3) selection evaluation depended on the utility function *U*, which included both potential and explicit factors; (4) the error distribution of each utility function followed a Gumbel distribution with an independent mean value of zero, while the error distribution of the rest of the stochastic actor functions followed a normal distribution; (5) independence of observations and no multicollinearity [33].

## 3. Data Collection and Analysis

Data were collected from the online questionnaire and were measured with a five point Likert scale (1: very negative to 5: very positive). A one-off survey method was adopted, and the respondents obtained a reward after completing the questionnaire. The contents of the questionnaire included

personal attributes, trip attributes, and attitude attributes of the respondents. The investigation was conducted from 19 December to 24 December 2018. In total, 519 questionnaires were sent and recovered. Samples in which respondents did not seriously answer the questions, samples with three or more questions void of answers, and samples with five continuous extreme values were eliminated. This left 452 questionnaires for analysis, with a validity rate of 87.09%.

## 3.1. Descriptive Statistical Analysis

In the valid survey questionnaires, the proportion of males and females in the sample was basically consistent. The respondents were mainly young people aged between 21 and 30 (70.4%). Occupations were mainly students and employees in enterprises, accounting for 68.4% of the total sample. Education levels were concentrated in bachelor degree or above (90%). Individuals earning less than CNY 2500 a month accounted for 30.1% of the sample, followed by CNY 7001 and 10,000 (19.9% total). The details are shown in Table 2.

**Table 2.** Demographic characteristics of the survey.

| Variables | | Frequency | Percentage | Variables | | Frequency | Percentage |
|---|---|---|---|---|---|---|---|
| Gender | Male | 239 | 52.9% | Education level | Senior high school and under | 16 | 3.5% |
| | Female | 213 | 47.1% | | Technical secondary school, junior college | 29 | 6.5% |
| Age | ≤20 | 4 | 0.9% | | University | 156 | 34.5% |
| | 21–30 | 318 | 70.4% | | Master degree or above | 251 | 55.5% |
| | 31–40 | 92 | 20.4% | | ≤2500 | 136 | 30.1% |
| | 41–50 | 30 | 6.6% | | 2501–4000 | 53 | 11.7% |
| | ≥51 | 8 | 1.7% | | 4001–5500 | 53 | 11.7% |
| | Student | 149 | 33.0% | | 5501–7000 | 54 | 11.9% |
| Occupation | Employee in enterprise | 160 | 35.4% | Income/ month | 7001–10000 | 90 | 19.9% |
| | Staff in government or public institution | 89 | 19.7% | | 10001–15000 | 46 | 10.2% |
| | Individual business | 9 | 2.0% | | >15000 | 20 | 4.5% |
| | Liberal professions | 14 | 3.1% | | | | |
| | Retirement | 7 | 1.5% | | | | |
| | Others | 24 | 5.3% | | | | |

Note: CNY 1000 ≈ USD 148.6.

## 3.2. Travel Characteristics Analysis

Among the most commonly used modes of transport, 54.9% of respondents chose express services, higher than the preference for taxis at 45.1%. Among the respondents who chose taxis as their trip mode, the proportion using car hailing software (63.2%) was higher than cruising (37.7%).

(1) The relationship between personal attributes and trip mode selection

It can be seen from Table 3 that the proportion of males choosing express services was higher (58.6%), while the percentage of females choosing taxis and express services was basically equivalent at 49.3% and 50.7%, respectively. In the relationship between age and travel mode choice, it can be seen that the proportion of young people choosing express services was obviously higher than those choosing taxis, but with the increase of age, the proportion choosing taxis increased gradually and eventually surpassed the proportion choosing express services. In terms of occupation, except among students, the proportion of business employees and other professions choosing express services was about 60%, and the proportion of other occupations choosing taxis was higher than the number choosing express services. The proportion of liberal professions and retirees choosing taxis was much higher than the proportion choosing express services. In terms of education level, except among those with a high school degree or below, the proportion of those who chose taxis was higher than the proportion choosing express services. Moreover, the proportion of those with other qualifications was

higher than that of those choosing taxi. For the factor of monthly income, people with incomes of CNY 2501–4000 and CNY 5501–7000 were more likely to choose taxis than express services.

**Table 3.** The relationship between personal attributes and trip mode selection.

| Personal Attributes | Modes | Taxi | | Express | |
|---|---|---|---|---|---|
| | | Frequency | Percentage | Frequency | Percentage |
| Gender | Male | 99 | 41.4% | 140 | 58.6% |
| | Female | 105 | 49.3% | 108 | 50.7% |
| Age | ≤20 | 1 | 25.0% | 3 | 75.0% |
| | 21–30 | 129 | 40.6% | 189 | 59.4% |
| | 31–40 | 48 | 52.2% | 44 | 47.8% |
| | 41–50 | 19 | 63.3% | 11 | 36.7% |
| | ≥51 | 7 | 87.5% | 1 | 12.5% |
| Occupation | Student | 60 | 40.3% | 89 | 59.7% |
| | Employee in enterprise | 66 | 41.3% | 94 | 58.8% |
| | Staff in government or public institution | 49 | 55.1% | 40 | 44.9% |
| | Individual business | 5 | 55.6% | 4 | 44.4% |
| | Liberal professions | 10 | 71.4% | 4 | 28.6% |
| | Retirement | 5 | 71.4% | 2 | 28.6% |
| | Others | 9 | 37.5% | 15 | 62.5% |
| Education level | Senior high school and under | 15 | 7.3% | 1 | 6.3% |
| | Technical secondary school, junior college | 11 | 37.9% | 18 | 62.1% |
| | University | 67 | 42.9% | 89 | 57.1% |
| | Master degree or above | 111 | 44.2% | 140 | 55.8% |
| Income/month | ≤2500 | 56 | 41.2% | 80 | 58.8% |
| | 2501–4000 | 30 | 56.6% | 23 | 43.4% |
| | 4001–5500 | 21 | 39.6% | 32 | 60.4% |
| | 5501–7000 | 29 | 53.7% | 25 | 46.3% |
| | 7001–10,000 | 39 | 43.3% | 51 | 56.7% |
| | 10,001–15,000 | 20 | 43.5% | 26 | 56.5% |
| | >15000 | 9 | 45.0% | 11 | 55.0% |

Note: CNY 1000 ≈ USD 148.6.

(2) The relationship between travel attributes and trip mode selection

It can be seen from Table 4 that the proportion of business offices using taxis was higher than the proportion of commuting and recreation users. For travel at night, the proportion choosing express services was obviously higher than those choosing taxis. Similarly, the percentage choosing express services was apparently higher than those choosing taxis when starting from suburban areas. In terms of travel distance, when the travel distance was less than 5 kilometers, express services were the main choice, while the proportion choosing taxis was higher when the travel distance exceeded 5 kilometers. When the waiting time was below 10 minutes, express services were the major travel mode.

**Table 4.** The relationship between travel attributes and trip mode selection.

| Modes / Travel Attributes | | Taxi | | Express | |
|---|---|---|---|---|---|
| | | Frequency | Percentage | Frequency | Percentage |
| Trip purpose | Commuting | 47 | 40.5% | 69 | 59.5% |
| | Recreation | 70 | 35.9% | 125 | 64.1% |
| | Business office | 71 | 63.4% | 41 | 36.6% |
| | Others | 16 | 55.2% | 13 | 44.8% |
| Travel time | Morning and evening peak time | 43 | 45.3% | 52 | 54.7% |
| | Mean time of daytime | 105 | 49.5% | 107 | 50.5% |
| | Night | 56 | 38.6% | 89 | 61.4% |
| Place of departure | Urban area | 178 | 50.6% | 174 | 49.4% |
| | Suburban area | 26 | 26.0% | 74 | 74.0% |
| Trip distance | <3 kilometers | 22 | 34.4% | 42 | 65.6% |
| | 3–5 kilometers | 62 | 38.3% | 100 | 61.7% |
| | >10 kilometers | 46 | 56.1% | 36 | 43.9% |
| Waiting time | <5 minutes | 90 | 45.2% | 109 | 54.8% |
| | 6–10 minutes | 85 | 42.9% | 113 | 57.1% |
| | 11–20 minutes | 23 | 53.5% | 20 | 46.5% |
| | >20 minutes | 6 | 50.0% | 6 | 50.0% |

### 3.3. Reliability Test

After obtaining the sample data through the formal questionnaire survey, a reliability test of the formal questionnaire was conducted again, and the Cronbach coefficient $\alpha$ of the seven latent variables involved in the model was calculated. The results are shown in Table 5. The reliability coefficient of each latent variable was greater than 0.8, indicating that the questionnaire had good reliability and could be used for structural equation analysis.

**Table 5.** Reliability test results.

| Number | Variables | Cronbach Coefficient ($\alpha$) | |
|---|---|---|---|
| | | Taxi | Express |
| 1 | Convenience (TC) | 0.898 | 0.933 |
| 2 | Safety (TS) | 0.942 | 0.923 |
| 3 | Reliability (TR) | 0.893 | 0.897 |
| 4 | Comfort (CC) | 0.931 | 0.939 |
| 5 | Economy (TE) | 0.838 | 0.805 |
| 6 | Perceived Value (PV) | 0.910 | 0.873 |
| 7 | Behavioral Intention (BI) | 0.892 | 0.894 |

## 4. Estimation Results

### 4.1. Parameter Estimation of SEM

It was necessary to calculate the adaptation coefficient of the latent variables before the establishment of the SEM-Logit model. To calculate the adaptation coefficient of the latent variables, the structural equation model needed to be established to estimate the path coefficient of each latent variable for the calculation of the adaptation coefficient. It was assumed that the travel behavior intention in the structural equation model was affected by the perceived value of the mode, and the perceived value was affected by convenience, reliability, comfort, safety, and economy. The structural equation model for establishing the choice behavior intention of an express service is shown in Figure 2. The load factor coefficients of each variable were at the 0.05 significance level. The specific test results are shown in Table 6, and the fitting results of the model are shown in Table 7.

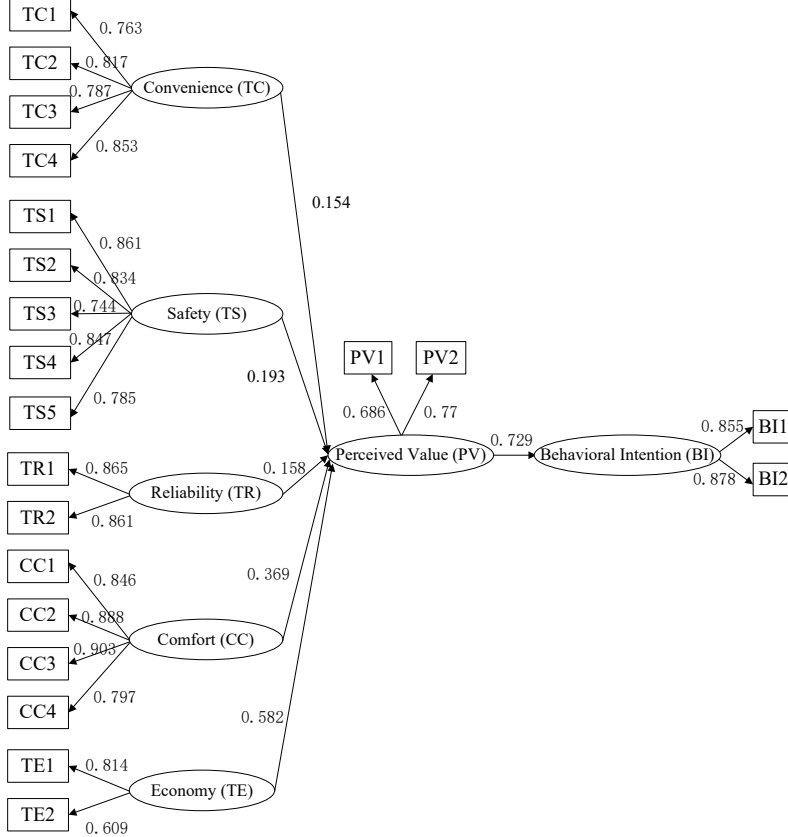

**Figure 2.** Standardized path coefficient of structural equation model (SEM) for express services.

**Table 6.** Load factor estimates and corresponding test values.

| Path | Taxi Model | | | | Express Model | | | |
|---|---|---|---|---|---|---|---|---|
| | Estimated Value | Standard Deviation | Critical Ratio | P | Estimated Value | Standard Deviation | Critical Ratio | P |
| PV <— TC | 0.172 | 0.034 | 3.977 | *** | 0.154 | 0.032 | 3.721 | *** |
| PV <— TS | 0.13 | 0.031 | 3.135 | 0.002 | 0.193 | 0.03 | 4.59 | *** |
| PV <— TR | 0.156 | 0.033 | 3.663 | *** | 0.158 | 0.028 | 3.744 | *** |
| PV <— CC | 0.411 | 0.035 | 8.775 | *** | 0.369 | 0.033 | 8.107 | *** |
| PV <— TE | 0.445 | 0.037 | 9.113 | *** | 0.582 | 0.041 | 10.688 | *** |
| BI <— PV | 0.826 | 0.067 | 14.195 | *** | 0.729 | 0.082 | 11.682 | *** |
| TC1<— TC | 0.773 | - | - | - | 0.763 | - | - | - |
| TC2 <—TC | 0.766 | 0.06 | 16.844 | *** | 0.817 | 0.057 | 18.555 | *** |
| TC3 <—TC | 0.782 | 0.056 | 17.278 | *** | 0.787 | 0.055 | 17.742 | *** |
| TC4 <—TC | 0.805 | 0.06 | 17.875 | *** | 0.853 | 0.054 | 19.563 | *** |
| TS1 <— TS | 0.899 | - | - | - | 0.861 | - | - | - |
| TS2 <— TS | 0.887 | 0.035 | 28.531 | *** | 0.834 | 0.043 | 22.629 | *** |
| TS3 <— TS | 0.792 | 0.043 | 22.492 | *** | 0.744 | 0.055 | 18.839 | *** |
| TS4 <— TS | 0.896 | 0.036 | 29.276 | *** | 0.847 | 0.046 | 23.251 | *** |
| TS5 <— TS | 0.77 | 0.046 | 21.355 | *** | 0.785 | 0.054 | 20.476 | *** |
| CC1<— CC | 0.846 | - | - | - | 0.846 | - | - | - |
| CC2<— CC | 0.886 | 0.042 | 24.714 | *** | 0.888 | 0.044 | 24.94 | *** |
| CC3<— CC | 0.837 | 0.044 | 22.387 | *** | 0.903 | 0.043 | 25.703 | *** |
| CC4<— CC | 0.785 | 0.043 | 20.203 | *** | 0.797 | 0.047 | 20.769 | *** |
| TR1<— TR | 0.78 | - | - | - | 0.865 | - | - | - |
| TR2<— TR | 0.872 | 0.051 | 20.197 | *** | 0.861 | 0.045 | 22.115 | *** |
| TE1 <— TE | 0.89 | - | - | - | 0.814 | - | - | - |
| TE2 <— TE | 0.647 | 0.054 | 14.686 | *** | 0.609 | 0.068 | 13.318 | *** |
| PV1<— PV | 0.712 | - | - | - | 0.686 | - | - | - |
| PV2<— PV | 0.865 | 0.063 | 17.154 | *** | 0.77 | 0.077 | 14.042 | *** |
| BI1 <— BI | 0.858 | - | - | - | 0.855 | - | - | - |
| BI2 <— BI | 0.856 | 0.05 | 19.454 | *** | 0.878 | 0.06 | 17.39 | *** |

**Table 7.** Fitness results.

| Evaluation Index | Test Results | | Fit Criteria |
|---|---|---|---|
| | Taxi Model | Express Model | |
| (Chi-Square/degrees of freedom ($\chi^2/df$) | 2.421 | 2.429 | 1–3 |
| Goodness-of-fit index (*GFI*) | 0.903 | 0.940 | >0.90 |
| Adjusted goodness-of-fit index (*AGFI*) | 0.877 | 0.905 | >0.90 |
| Root mean square residual (*RMR*) | 0.034 | 0.027 | <0.05 |
| Root mean square error ofapproximation (*RMSEA*) | 0.073 | 0.073 | <0.08 |
| Normed fit index (*NFI*) | 0.910 | 0.901 | >0.90 |
| Incremental fit index (*IFI*) | 0.935 | 0.928 | >0.90 |
| Comparative fit index (*CFI*) | 0.934 | 0.927 | >0.90 |

*4.2. Parameter Estimation of the SEM-Logit Model*

The trip mode choice behavior of taxi and express services considering the latent variables was based on the BL (binary logit) model, and the five latent variables of perceived value—convenience, safety, reliability, comfort, and economy—were substituted into the choice model. Afterwards, software was used to calibrate and verify the model parameters. Taking taxis as a reference, the utility function of the express services could be established as:

$$
\begin{aligned}
V_{express} &= Constant + \theta_1 \cdot Gender + \theta_2 \cdot Age + \theta_3 \cdot Occupation + \theta_4 \cdot Education \\
&+ \theta_5 \cdot Income + \theta_6 \cdot Purpose + \theta_7 \cdot Time + \theta_8 \cdot Lacation + \theta_9 \cdot Distance \\
&+ \theta_{10} \cdot Waiting + \theta_{11} \cdot TC + \theta_{12} \cdot TS + \theta_{13} \cdot TR + \theta_{14} \cdot CC + \theta_{15} \cdot TE
\end{aligned}
\tag{6}
$$

Using the estimated path coefficient and the corresponding structural equation model, adaptation coefficients of the latent variables of perceived attitude, such as convenience, safety, reliability, comfort, and economy, were substituted into the choice model. These latent variables, personal attributes, and the characteristic vectors of trip attributes were taken as the characteristic variables of trip utility to calibrate and verify the model parameters. Taking taxis as a reference level, the model parameter estimation results were obtained according to the basic attribute survey data and the survey data of attitude perception. The specific parameter estimation results and relevant test results are shown in Table 8.

**Table 8.** Parameter estimation and test.

| Variables | Value | S.E. | Sig. | Likelihood Ratio (Exp(B)) |
|---|---|---|---|---|
| Gender | 0.276 | 0.234 | 0.238 | 1.318 |
| Age | −0.493 | 0.195 | 0.011 | 0.611 |
| Occupation | −0.008 | 0.085 | 0.299 | 0.992 |
| Education | 0.119 | 0.176 | 0.052 | 1.127 |
| Income | 0.111 | 0.064 | 0.044 | 1.118 |
| Purpose | −0.403 | 0.136 | 0.003 | 0.668 |
| Time | 0.196 | 0.159 | 0.118 | 1.216 |
| Location | 1.290 | 0.305 | 0.000 | 3.632 |
| Distance | −0.245 | 0.125 | 0.051 | 0.783 |
| Waiting | 0.010 | 0.161 | 0.548 | 1.010 |
| TC | 1.678 | 0.286 | 0.000 | 5.355 |
| TS | −0.037 | 0.219 | 0.565 | 0.964 |
| TR | −0.091 | 0.223 | 0.283 | 0.913 |
| CC | −0.280 | 0.237 | 0.038 | 0.756 |
| TE | 0.854 | 0.223 | 0.000 | 2.350 |
| Constant | −5.250 | 1.067 | 0.000 | 0.005 |

According to the parameter calibration results from Table 8, the trip mode choice model could be obtained, and the selection function is shown below:

$$\ln \frac{P_{1n}}{P_{0n}} = -5.250 + 0.276 Gender - 0.493 Age - 0.008 Occupation$$
$$+0.119 Education + 0.111 Income - 0.403 Purpose + 0.196 Time$$
$$+1.290 Lacation - 0.245 Distance + 0.010 Waiting + 1.678 TC \tag{7}$$
$$-0.037 TS - 0.091 TR - 0.280 CC + 0.854 TE$$

where $P_{0n}$ is the probability that passengers choose a taxi, and $P_{1n}$ is the probability that passengers choose an express service.

The test of the model mainly included the chi-square ($\chi^2$) test of the likelihood ratio, the McFadden test ($\rho^2$), and the hit rate test.

(1) Under the hypothesis $H_0$: $\theta_1 = \theta_2 = \ldots = \theta_k = 0$, and at the significance level of 5%, the chi-square test value $\chi_\alpha^2$ of 15 degrees of freedom is 24.996, when $-2(L(0) - L(\hat{\theta})) = 479.404 > \chi_\alpha^2$ and the null hypothesis is rejected, which indicates that the feature vector of the model has a significant impact on the choice of trip mode.

(2) The McFadden's determination coefficient $\rho^2$ of the model is 0.271, which is higher than the BL model without considering the latent variables ($\rho^2 = 0.116$), indicating that the precision of the model when considering the latent variables has better precision.

(3) As can be seen from Table 9, the hit ratio of the SEM-Logit model is higher than that of the BL model.

**Table 9.** Hit ratio test.

| Model | Hit Rate | Overall Hit Ratio | Hit Rate per Mode |
|---|---|---|---|
| SEM-Logit model | Taxi | 73.0% | 66.7% |
| | Express | | 78.2% |
| BL model | Taxi | 63.3% | 52.9% |
| | Express | | 71.8% |

## 5. Discussions and Conclusions

According to the estimated values of the characteristic variables in Table 8, it can be seen that latent variable factors had significant impacts on the choice results. Single-factor analysis was conducted on travelers' trip mode choice behaviors. Assuming that other characteristic variables remained invariable, the impact of changing a certain characteristic variable on the choice of travel mode was quantified.

(1) Personal attributes

In variables relating to personal attributes, the significant variables were age, education level, and income. The regression coefficient of age in the model was −0.493, indicating that, with the increase of age, travelers were more inclined to take taxis, which is also consistent with the commonly held belief that smartphones are less popular among older travelers. The regression coefficient of education level in the model was 0.119, with a probability ratio of choosing taxis and express services among passengers of 1.127, indicating that, in the survey, with the improvement of degree education to the next level, the passengers choosing express services increased 1.127 times. The regression coefficient of income was positive, showing that, with an increase of income, passengers preferred to choose express services.

(2) Travel attributes

In variables relating to travel attributes, the travel time and the waiting time were not significant. The regression coefficient of travel purpose was −0.403, showing that, in the travel mode choice of non-commuters, people were more inclined to choose taxis. The regression coefficient of departure place was 1.290, indicating that passengers taking rides from suburban areas preferred to choose online car-hailing. This is consistent with the fact that taxis are mostly concentrated in urban areas, and few are in suburban areas. The regression coefficient of travel distance was negative, and the probability

ratio was 0.783, meaning that, for long-distance travel, passengers were more inclined to use taxis due to the fact that most long-distance travel was business trips, and the probability of using taxis by business passengers was higher.

(3)    Attitude perception attribute

In variables relating to attitude perception attributes, convenience, comfort, and economy were significant. Reliability and safety did not pass the significance test. The regression coefficient of convenience was 1.678, indicating the number of passengers who thought the convenience of express services was higher than that of taxis, with a probability ratio of 5.355. That is, when the convenience perception of passengers improved by one level, the probability of choosing express services increased by 5.355 times. Similarly, the regression coefficient of comfort was −0.280, showing that passengers were not satisfied with the comfort of express services. The regression coefficient of economy was 0.854, showing that passengers were satisfied with the economy of express services.

This paper selected taxis and express services as the investigated modes of travel. The structural equation model and the factor analysis method were used to calculate the fitted values of the latent variables. The calculation results were substituted into the utility function and the established trip mode choice behavior of taxis and express services considering the latent variables to quantitatively describe the impacts of latent variables on choice results. The results showed that the SEM-Logit model including latent variables was better than the BL model without consideration of latent variables in terms of model precision and hit ratio. Meanwhile, after introducing the latent variables, it was found that convenience, comfort, and economy had a significant influence on the model, and the explanatory power of the model increased accordingly.

**Author Contributions:** The individual responsibilities and contribution of the authors are listed as follows: Y.S. designed the research, developed the model and wrote the paper; H.G. guided the research process and revised the manuscript; Y.C. conducted the model validation helped edit the manuscript.

**Acknowledgments:** This research was supported by the National Natural Science Foundation of China (Grant No.51338008 and 51378036).

**Conflicts of Interest:** The authors declare no conflict of interest.

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
