# Peer review of "Research on the Choice Behavior of Taxis and Express Services Based on the SEM-Logit Model"

_sustainability, doi:10.3390/su11102974_

Reviewer 1 Report

Not much to tell to this interesting manuscript. Congrats to the authors! Just check these two issues.

-        the observation that researches on choice behaviors of taxi and express modes is unusual in scientific literature, so the paper, when published, might certainly advance knowledge in this field

-        the methodology is well reported, although authors could have explained why SEM Logit is applied against other methods, and possibly expanded further sections 2.2 on hypotheses. However, scientific soundness is a merit of this manuscript.

-        the case study is interesting, for location, size and outcomes

-        and scientific literature references to corroborate all of the above is appropriate.

Aside from the typo in line 83 and the suggestion to convert Chinese currency into some others to facilitate non-Chinese readership, I do not see further areas of improvement, aside from language check, maybe (I’m no English-native, but to me it was readable).

a description of the differences between taxis and on-line hailing service operations could help readers non familiar with paratransit operations, but this is really a minor issue.

r 84 check text to remove or adjust the reference

r 204 report alternative currency as well, e.g USD or Euros

Author Response

Dear reviewer:

 Thanks for your evaluation and affirmation of my paper. According to your suggestions, I have revised the manuscript and made the explanations as following:

Point 1: A description of the differences between taxis and on-line hailing service operations could help readers non familiar with paratransit operations.

Response 1: In the second paragraph of Introduction, a description of “Unlike taxi, online-hailing services can only carry passengers through network appointment and are not allowed to cruise.” was stated

Point 2: r 84 check text to remove or adjust the reference

Response 2: The reference was adjusted to “Zhang (2017) [17] used Push-Pull-Mooring (PPM) model to study the transfer intention of online car-hailing users.”

Point 3: r 204 report alternative currency as well, e.g USD or Euros

Response 3: A rate description “(Note: CNY1000USD148.6)” was added below Tables 2 and 3 respectively.

Reviewer 2 Report

The paper try to understand passenger mode choice between taxi and online car service with an SEM-Logit model. Model precision is compared between SEM-Logit and general logit model result to indicate that precision is improved by including latent variables. The paper found that convenience, comfort and economy factors are significant contributors.

The manuscript contains some grammar issues which make the paper really hard to read and very annoying, suggest to do a thorough editorial review by a professional spell and grammar editor. Few examples, Line 38, “service indicators characteristics” should be “service-indicator characteristics”. Line 39, “not obvious as taxi and subway or bus” should be “not as obvious as taxi to subway or taxi to bus”. Line 84, take care of “Error!” message. But similar issues exist throughout the whole paper. A professional editorial review is recommended for the whole paper.

The manuscript has its merits to the literature. The reviewer appreciate the latent variable contributions to transit mode choice however, the paper need to address the following major concerns from the reviewer.

Line 38 to 39, in this paragraph, the authors stated that online-car hailing and taxi service has very similar service characteristics and hard to distinguish for mode choice. The reviewer feel the statement is not clear if the authors did not define or clarify what the online car-hailing and taxi service are. Clarifications are recommended. Is online car-hailing service including ride-hailing app requested taxi service? Is the taxi service including street hailing and phone-call reservations?  To me all the door-to-door transit service can be categorized as on-demand mobility service, on-demand shared mobility service, traditional street hailing taxi service, and dispatched taxi service. For dispatched taxi service, it contains on-line booking and phone-call reservation. Please include the following paper in your literature reference and address door-to-door mobility services that you referred as research modes in your paper.

Raj Bridgelall, Pan Lu, Denver Tolliver, and Tie Xu. Mining Connected Vehicle Data for Beneficial Patterns in Dubai Taxi Operations, Journal of Advanced Transportation, Volume 2018, Article ID 8963234, 8 pages, 2018.

Line 147, section 2.2, the authors discussed a few assumptions about the models. However, the reviewer felt the authors missed few model assumptions: 1) independent of observations and 2) no multicollinearity

Please refer to the following newly published paper and cite it in your research, Pan L., Hao W., and Denver T., Prediction of Bridge Components Condition Ratings Using Ordinal Logistic Regression Model, Mathematical Problems in Engineering, Volume 2019, Article ID 9797584, 2019. The paper clearly indicates that Yi, the dependent observations should be statistically independent with each other. The reviewer feel that authors need to address this two assumptions with the data and the model results to indicate the significance of the model results and their interpretations.

Line 163, the authors define the two mode choices as taxi vs express. And again, here need to clearly define what are they. Is taxi including taxi company dispatched service? In other words, on-line booking and phone-call reservation?

This is critical since the authors try to capture the mode choice for door-to-door mobility service, if the choices are not exclusively complete then the results are biased. In other words, a person not choose “taxi” does not necessary will choose “express” may choose other alternative door-to-door service modes.  How the questionnaire is designed is also need to clarify to help readers to better understand how the mode choice is determined.

Another general comment is related to how the authors address two modes, please keep it consistent.  Both online car service and express were used in the paper and is very confusing.

Author Response

Dear reviewer:

Thanks for your evaluation and affirmation of my paper. According to your suggestions, I have revised the manuscript and made the explanations as following:

Point 1: The manuscript contains some grammar issues which make the paper really hard to read and very annoying, suggest to do a thorough editorial review by a professional spell and grammar editor. Few examples, Line 38, “service indicators characteristics” should be “service-indicator characteristics”. Line 39, “not obvious as taxi and subway or bus” should be “not as obvious as taxi to subway or taxi to bus”. Line 84, take care of “Error!” message. But similar issues exist throughout the whole paper. A professional editorial review is recommended for the whole paper.

Response 1: The original manuscript has been revised according to the proposal.

Point 2: Line 38 to 39, in this paragraph, the authors stated that online-car hailing and taxi service has very similar service characteristics and hard to distinguish for mode choice. The reviewer feel the statement is not clear if the authors did not define or clarify what the online car-hailing and taxi service are. Clarifications are recommended. Is online car-hailing service including ride-hailing app requested taxi service? Is the taxi service including street hailing and phone-call reservations?  To me all the door-to-door transit service can be categorized as on-demand mobility service, on-demand shared mobility service, traditional street hailing taxi service, and dispatched taxi service. For dispatched taxi service, it contains on-line booking and phone-call reservation. Please include the following paper in your literature reference and address door-to-door mobility services that you referred as research modes in your paper.

Response 2: In the second paragraph of Introduction, a description of “Unlike taxi, online-hailing services can only carry passengers through network appointment and are not allowed to cruise.” was stated. In Introduction section, the Figure 1 was added to illustrate the categories of taxi and cited Bridgelall R, Lu P, Tolliver D D, et al. Mining Connected Vehicle Data for Beneficial Patterns in Dubai Taxi Operations[J]. Journal of Advanced Transportation, 2018, 2018.” as the reference.

Point 3: Line 147, section 2.2, the authors discussed a few assumptions about the models. However, the reviewer felt the authors missed few model assumptions: 1) independent of observations and 2) no multicollinearity

Response 3: In section 2.2, the assumptions of “(5) Independent of observations and no multicollinearity.” was added and corresponding reference citations.

Point 4: Line 163, the authors define the two mode choices as taxi vs express. And again, here need to clearly define what are they. Is taxi including taxi company dispatched service? In other words, on-line booking and phone-call reservation?

Response 4: The difference between the two service modes was illustrated in Figure 1.

Point 5: This is critical since the authors try to capture the mode choice for door-to-door mobility service, if the choices are not exclusively complete then the results are biased. In other words, a person not choose “taxi” does not necessary will choose “express may choose other alternative door-to-door service modes.  How the questionnaire is designed is also need to clarify to help readers to better understand how the mode choice is determined.

Another general comment is related to how the authors address two modes, please keep it consistent.  Both online car service and express were used in the paper and is very confusing.

Response 5: It is true that this situation exists. The reason why this paper only discusses the two modes of Taxi and Express is mainly because: (1) Highly similar service modes existed in the two travel modes. (2) The research focused on the influence mechanism of latent variables on the selection results among the travel modes with highly similar service modes.

Reviewer 3 Report

General comments: The paper is an attempt to report analysis of stated survey of choosing two transport services: taxi and online-car hailing (e.g. uber). Comparing the analysis to exhibit proposition of Binari Logit vs Latent-variables logit models. The results show estimation comparisons and conclude with superiority of latent-variables model. There are interesting elements of the study that possibly merit publication but the current state of paper is not accessible to much extent.

Major concerns:

·         The rationale of understanding travel behaviour is interesting of online-car hailing is interesting but this is not supported with proper literature review. I am very surprise to read that choice behaviour which result from many factors acted together is rarely been studied. I as the authors used McFadden, Ben-Akiva, Morikawa and Spear works dated from a long time ago. I guess there would be loads of studies since then which are not captured in this paper significantly.

·         I am confused as to why the authors start with taxi vs express research questions and ended up with SEM-logit vs BL conclusion. Please clarify.

·         Methodology section also need a much clearer narration on how adopted methods are appropriate. Do we need to see all the equations? What recent literature inform the development of Table 1? The ones used are quite old, new references are needed.

·         Very little written about the data collection framework. Is this a one-off survey or part of a bigger commissioned study which normally will have seminal report which should be made clear in the paper, so readers are well informed of how reliable the study is.

·         Overall the paper is still too technical and very little to take for any meaningful theoretical as well as practical use. Contribution for research implication and managerial practice should be made clearer.  

·         Many typo errors: table number with the text are not always consistent. Equation model also use incorrect coefficient, e.g. equation (7) on education and occupation coefficients.

·         Some literature are not accessible: I can’t find ref no 1 in any source? Validity issue!

Author Response

Dear reviewer:

Thanks for your evaluation and affirmation of my paper. According to your suggestions, I have revised the manuscript and made the explanations as following:

Point 1: The rationale of understanding travel behaviour is interesting of online-car hailing is interesting but this is not supported with proper literature review. I am very surprise to read that choice behaviour which result from many factors acted together is rarely been studied. I as the authors used McFadden, Ben-Akiva, Morikawa and Spear works dated from a long time ago. I guess there would be loads of studies since then which are not captured in this paper significantly.

Response 1: This paper focuses on the influence of latent variable factors on the selection results, so literatures in the methodology are mainly related to the discrete selection models considering latent variable factors.

Point 2: I am confused as to why the authors start with taxi vs express research questions and ended up with SEM-logit vs BL conclusion. Please clarify.

Response 2: The problem discussed in this paper is the choice of taxi and express with similar travel characteristics and service characteristics. The method used in this paper is SEM-Logit method. To illustrate the effectiveness of the method, the accuracy of SEM-Logit model and BL model is compared in the last part.

Point 3: Methodology section also need a much clearer narration on how adopted methods are appropriate. Do we need to see all the equations? What recent literature inform the development of Table 1? The ones used are quite old, new references are needed.

Response 3: A model description was added in Section 2.1. “The model consists of two parts: the first part is SEM model, which is mainly used to describe the causal relationship between latent variables of travel mode selection and the corresponding observation variables; The second part is the Logit model, which is used to express the nonlinear function relationship between the probability of choosing a certain travel plan and the potential variables influencing the decision.” Literatures [27] and [33] were added in the description of latent variable selection.

Point 4: Very little written about the data collection framework. Is this a one-off survey or part of a bigger commissioned study which normally will have seminal report which should be made clear in the paper, so readers are well informed of how reliable the study is.

Response 4: In section 3, a description of “The one-off survey method was adopted and respondents can get a certain amount of rewards after completing the questionnaire.” was stated.

Point 5: Overall the paper is still too technical and very little to take for any meaningful theoretical as well as practical use. Contribution for research implication and managerial practice should be made clearer.

Response 5: The explanation of theoretical and practical significance was added In Introduction: “For the theoretical aspect, this paper conducted the model analysis of selection behavior of taxi drivers and passengers in psychological level, a more suitable mechanism analysis of behavior selection model was put forward. In practice, the survey data was used for calibrating the model, which can provide suggestions for relevant departments to develop the sustainable development policy of the taxi and on-line car hailing, optimize scheduling distribution, as well as improve the passenger satisfaction.”

Point 6: Many typo errors: table number with the text are not always consistent. Equation model also use incorrect coefficient, e.g. equation (7) on education and occupation coefficients.

Response 6: Table 6”in “According to the parameter calibration results from Table 6”of  section 3.4 was adjusted to “Table 8”. The incorrect coefficient in equation (7) was corrected.

Point 7: Some literature are not accessible: I can’t find ref no 1 in any source? Validity issue!

Response 7: ref no 1 is a Chinese literature,  it is available on the website of http://en.cnki.com.cn/Article_en/CJFDTotal-JTBH201805003.htm

Reviewer 4 Report

The paper investigates a very important scientific issue; the role of travel attributes in travel choice. It has to be stressed that there are several influencing factors of travel choice and some "soft" factors such as feelings and perceived values are also considered. The applied Logit model and SEM-Logit model are suitable for this analysis and the conducted survey with 452 valid answers is appropriate for drawing coclusions.

However, before publication some modifications have to be made from my point of view.

First, the sustainability issue should be more stressed at least in the Introduction section. The competitor trip mode of taxi is public transport which is a more environmental friendly form of travel and the choice criteria are referred here as well. Some explanaition would be necessary about the trade-off between taxi and public mode from the perspective of attributes.

Second, for the SEM -logit model it has to be mentioned that there are other suitable methods for determining the weights of the attributes, e.g. MCDM methods and within that, AHP. I suggest amending the Methodology section by this statement, supported by providing some references.

There are some tiny mistakes to be corrected:

Line 84. the reference for Zhang (2017) is not numbered.

Line 164. instead of "including" I suggest "includes".

Recommended references:

Duleba et al (2012): A dynamic analysis on public bus transport's supply quality by using AHP. Trabnsport, 27(3), 268-275

Ghorbanzadeh et al (2018): Sustainable urban transport planning considering different stakeholder groups by an Interval-AHP decision support model. Sustainability, 11(1), 1-18.

Huang and Shuai (2018): A methodology for calculating the passenger comfort benefits of railway travel. Journal of modern transportation, 1-12.

Duleba and Moslem (2018): Sustainable urban transport development with stakeholder participation, an AHP-Kendall model: A case study for Mersin. Sustainability, 10(10), 3647. 

Author Response

Dear reviewer:

Thanks for your evaluation and affirmation of my paper. According to your suggestions, I have revised the manuscript and made the explanations as following:

Point 1: First, the sustainability issue should be more stressed at least in the Introduction section. The competitor trip mode of taxi is public transport which is a more environmental friendly form of travel and the choice criteria are referred here as well. Some explanaition would be necessary about the trade-off between taxi and public mode from the perspective of attributes.

Response 1: Add description in Introduction section: Online car-hailing service realizes the information matching of driver and passenger, and brings down the proportion of customer searching. Meanwhile, it can replace the private car travel needs of some high time value groups. Hence, it is a sustainable travel mode.

Point 2: Second, for the SEM -logit model it has to be mentioned that there are other suitable methods for determining the weights of the attributes, e.g. MCDM methods and within that, AHP. I suggest amending the Methodology section by this statement, supported by providing some references.

Response 2: Since this main purpose of this paper is to build a quantitative selection model and not only consider the weight of influencing factors, so the discrete selection model is only considered in this paper.

Point 3: There are some tiny mistakes to be corrected:

Line 84. the reference for Zhang (2017) is not numbered.

Line 164. instead of "including" I suggest "includes".

Response 3: The reference number for Zhang (2017) is added and "including" is modified to "includes".

Round  2

Reviewer 2 Report

The authors revised the manuscript and addressed all my earlier recommendations with my satisfaction. Thanks.

Author Response

Dear reviewer:

Thanks for your evaluation and affirmation of my paper.

Point 1: The authors revised the manuscript and addressed all my earlier recommendations with my satisfaction. Thanks.

Response 1: Thanks!

Reviewer 3 Report

Please delete the last sentence of the first paragraph in introduction section.

Author Response

Dear reviewer:

Thanks for your evaluation and affirmation of my paper.

Point 1: Please delete the last sentence of the first paragraph in introduction section.

Response 1: This sentence “Choice behavior is the result of many factors acting together, and has rarely been studied.” has been deleted.